# Calculation of minimum energy pathways in transport proteins
Briony A. Yorke [1] & Helen M. Ginn [2,3] ✉

Although static structures of protein metastable states are well-studied, the fleeting transitions between these states are difficult to experimentally observe or predict. We present a computationally inexpensive algorithm, "cold-inbetweening", which generates trajectories between experimentally determined end-states. Here we apply cold-inbetweening to provide mechanistic insight into the ubiquitous alternate access model of operation in three membrane transporter superfamilies. Here, we study DraNramp from *Deinococcus radiodurans*, MalT from *Bacillus cereus*, and MATE from *Pyrococcus furiosus*. In MalT, the trajectory demonstrates elevator transport through unwinding of a supporter arm helix, maintaining adequate space to transport maltose. In DraNramp, outward-gate closure occurs prior to inward-gate opening, in accordance with the alternate access hypothesis. In the MATE transporter, switching conformation involves obligatory rewinding of the N-terminal helix to avoid steric backbone clashes. This concurrently plugs the cavernous ligand-binding site mid-conformational change. Cold-inbetweening can generate hypotheses about large functionally relevant protein conformational changes.

Understanding transitions between conformational states in proteins is essential to fully characterise their mechanisms. However, large transitions present a significant challenge in structural biology, as they involve rapid, infrequent energy barrier crossing events that occur stochastically. These transitions can be orders of magnitude shorter than the lifetimes of intermediate metastable states, making them difficult to study experimentally or computationally. Molecular dynamics (MD) simulations can provide information about intermediate states and transitions, but such simulations are computationally expensive. Reducing protein models to a series of springs (elastic network models)[1–3] or grouping atoms into beads (coarse-grained models)[4,5], significantly reduces the computational costs of simulating large transitions. However, both of these methods lack the resolution necessary to elucidate molecular mechanisms, such as specific interactions between amino acid side chains[1].

Alternatively, a wide range of accelerated MD approaches exist, each employing specialised sampling or biasing algorithms to increase the frequency of barrier crossing and expand the accessible conformational space. Biased sampling techniques deliberately alter the potential energy landscape or apply external forces to encourage transitions between states, allowing rare events to occur on computationally accessible timescales. Examples include umbrella sampling[6,7], which applies harmonic restraints along a chosen reaction coordinate: metadynamics, which deposits a history-dependent bias to discourage revisiting explored states[8,9], and steered MD,

which applies directional forces to induce desired conformational changes[10,11].

Unbiased adaptive sampling approaches, such as replica exchange MD[12,13] or high-throughput iterative simulations[14,15], instead rely on running multiple trajectories under different conditions to explore conformational space more efficiently without modifying the energy landscape. Biased methods can accelerate sampling but careful choice of sampling strength, window size, placement and overlap are needed to reduce the risk of distorting the true free energy landscape[16], while unbiased strategies preserve physical realism but often require far greater computational resources to capture rare events.

Markov state models (MSMs) reconstruct long-timescale dynamics from ensembles of shorter MD trajectories[17]. Unlike biased sampling approaches, MSMs preserve native dynamics and do not artificially accelerate transitions. They organise calculated conformations into discrete states and estimate the transition probabilities between them. When combined with unbiased adaptive sampling strategies, MSMs can guide the generation of new trajectories by identifying undersampled regions of conformational space, providing both thermodynamic and kinetic insight without introducing artificial bias. MSMs rely on having a sufficiently large set of sampled trajectories, which can be limiting for large transitions in complex proteins[18].

Transition path sampling (TPS) focuses specifically on the rare transition events between metastable states. By generating an ensemble of

[1]School of Chemistry and Astbury Centre, University of Leeds, Leeds, UK. [2]Center for Free-Electron Laser Science CFEL, Deutsches Elektronen-Synchrotron DESY, Notkestraße 85, 22607 Hamburg, Germany. [3]Institute for Nanostructure and Solid State Physics, University of Hamburg, Hamburg, Germany. ✉e-mail: helen.ginn@cfel.de

trajectories using Monte Carlo moves in trajectory space, TPS captures transitions without wasting computational resources simulating comparatively long-lived metastable states. Various TPS approaches exist with bottlenecks arising during either trajectory calculation or the evaluation of the probability of a trajectory. Inefficiency arises where the majority of trajectories generated are improbable[19]. TPS is the most suitable MD-based technique for simulating an ensemble of transitions. It remains a computationally and energy-intensive procedure[20,21] an issue not mitigated by recent implementations of machine learning algorithms and deep learning approaches[22,23].

Techniques based on molecular dynamics employ a Markov model where the positions and forces on each atom determine the following timepoint and a history of the trajectory is not required. Models outside of this paradigm have also been developed for interpolating between computationally or experimentally-determined endpoints[24–28]. Some are considered aesthetical rather than analytical tools[29]. In some cases, implementations of the algorithm are no longer available, and many of the methods were deliberately designed as a demonstrative tool rather than providing any testable hypothesis of protein motion. All of these methods require the researcher to supply two trusted structures to serve as endpoints, and from these, interpolated structures are hypothesised.

Linearly interpolated paths may be optimised using the nudged elastic band (NEB)[30] or string method[31] to construct minimum energy pathways between two conformations. In NEB, intermediate structures are generated and connected with virtual springs to form a so-called elastic band. The forces are then iteratively minimized so that the structures relax onto the minimum energy pathway. Although NEB does not require full MD simulation, it is limited by sensitivity to the number of intermediate structures available and there are few examples of its application to the study of proteins[32]. Conversely the string method works by representing the transition as a continuous curve of intermediate conformations, which is iteratively relaxed under the system's forces to converge on the minimum energy pathway. Path optimisation by the string method is more widely used for proteins but is computationally more challenging due to the requirement of multiple short MD runs to estimate mean forces[33].

Additional approaches based on normal mode analysis[34,35] have been successful for generating hypotheses for site-specific transitions but are inappropriate for large-scale structural transitions because they rely on harmonic approximations around a single energy minimum and are therefore unable to capture the non-linear, anharmonic nature of large conformational changes. Due to these limitations, the way to and from experimentally determined end-point structures remain under-explored.

To address this, we have developed a computationally inexpensive algorithm outside of the MD paradigm, cold-inbetweening, that generates trajectories which smoothly connect starting and ending conformations. Inbetweening is a traditional method in animation for filling in gaps between keyframes. We refer to this as 'cold' inbetweening because we deliberately omit the model of heat.

The algorithm mimics the nature of protein flexibility by allowing rotation around bonds. According to the equipartition theorem, energy terms should be equally distributed between bond stretching, angle bending and rotations. However, due to the much higher magnitude of forces involved in altering bond lengths and angles in comparison to rotations around the bond, the torsion angle changes are far more significant for large conformational changes in protein structure. Therefore, due to the simplification of the parameter space to include only torsion angles, larger conformational changes can be studied at a lower computational cost than other methods. The algorithm is designed to minimise fluctuations in kinetic and potential energy needed to complete a transition between experimentally determined end-states. The model of motion omits random thermal fluctuations other than those associated with the desired conformational change.

To test the cold-inbetweening algorithm we applied it to the analysis of transport protein mechanisms from three distinct superfamilies to investigate whether the calculated trajectories were in agreement with the evidence

in the literature. While these mechanisms were chosen to demonstrate the method, it is widely applicable to any protein where high quality and fully modelled starting and ending structures are available.

Transport proteins are able to move specific molecules across a membrane with or against a concentration gradient. Biochemical data demonstrated that membrane transporters could transport a cognate ligand strictly over non-cognate solutes[36]. This led to the development of the "alternate access hypothesis". This hypothesis states that the transporter binding site is exposed at one side of a membrane at a time, allowing it to non-covalently bind a specific ligand, without admitting other solutes or solvent.

Further hypotheses were introduced to discuss how this may occur mechanistically[37]. The binding event was thought to initiate a conformational change that translocates the ligand through the membrane. The binding site is then subsequently exposed to the other side of the membrane and the ligand released. This hypothesis was further refined as structural evidence became available, well-reviewed in ref. 38. Some energetically unfavourable modes of transport were ruled out, such as a travelling ligand-binding site from one side of the membrane to the other. The remaining hypotheses are described in terms of the behaviour of two "bundles" and one "barrier". Here, "bundle" refers to either a domain or a subset of a domain, which can be considered to either exhibit concerted motion or no motion as a fixed unit. The "barrier" refers to the close proteinaceous contact of the two bundles, which prevents free solvent and solute flow through the transporter. The barrier itself is defined as either "fixed" (negligible translocation of the barrier perpendicular to the membrane) or "moving" (undergoes conformational changes which shifts the barrier towards either side of the membrane).

The alternate access mechanisms are distilled into three major hypotheses[39]. The "elevator mechanism" involves a fixed barrier, with the first bundle fixed in place relative to the membrane, while the second bundle makes the most extensive interactions with the ligand and moves to transport it from one side to the other. We use the EIIC domain of MalT as an exemplar of the elevator mechanism[40] (Fig. 1A). The "rocking-bundle" and "rocker-switch" mechanisms are the two moving barrier mechanisms. Here, the bundles undergo a conformational change to move the barrier around the ligand, to expose it to either side of the membrane at each extreme of the overall motion. The difference between them is that, for a rocking-bundle, the vast majority of this conformational change is achieved by one bundle, whereas for a rocker-switch, this conformational change is shared between both bundles. We use the bacterial manganese transporter DraNramp as an example of a rocking-bundle protein (Fig. 1B), and the MATE transporter as an example of a rocker-switch protein (Fig. 1C).

Transport proteins are particularly suited to our method since they often have a few dominant conformations: inward-open (where the binding site is exposed to the inner side of the membrane), outward-open (where the binding site is exposed to the outer side of the membrane) and occluded (where the binding site is shielded from both sides of the membrane). However, the details of the motion involved in these mechanisms are unknown. We selected three classes of conformational change to test the cold-inbetweening algorithm. Specific transport proteins were chosen according to the availability of high-quality models of the inward-open and outward-open states in the Protein Data Bank (PDB) to use as our starting and ending conformations. We demonstrate that the cold-inbetweening algorithm be used to investigate the mechanism of action.

## Results

The algorithm assumes regularised bond lengths, angles and optimised torsion angles[41] of both starting and ending structures which should be supplied as PDB files with explicitly modelled hydrogen atoms. Regularisation is automatically taken care of in the process of importing the structures in the RoPE GUI where cold-inbetweening is implemented[42]. The GUI also allows for export of paths in PDB format, editing and visualisation of the pathway using a draggable slider. Close pairs of atoms (separated by at least three bonds, less than 8 Å and not part of an aromatic ring) in the

**Fig. 1 | Visual representation of the three hypothesised types of alternate access mechanisms.**
**A** the elevator mechanism, as exhibited by MalT, **B** the rocking-bundle mechanism, as demonstrated by DraNramp, and **C** the rocker-switch mechanism, as exhibited in MATE. The two bundles are coloured in mauve, and the barrier between them is coloured in a dark mauve.

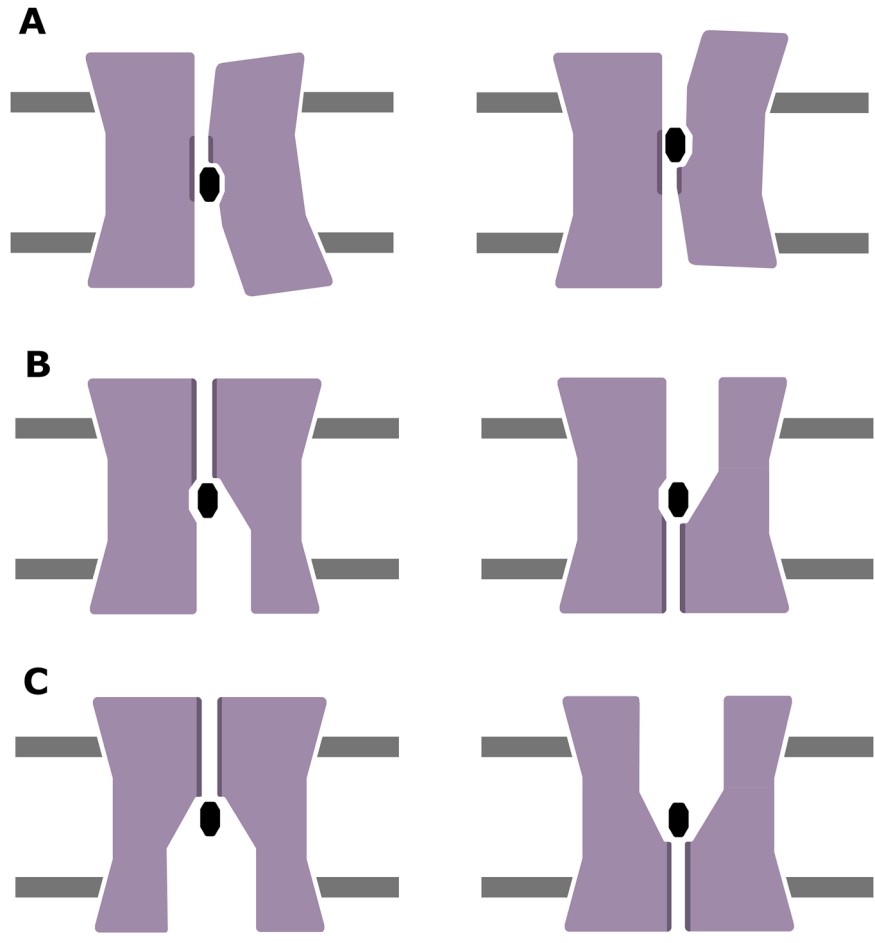

**Fig. 2 | Description of the parameters and target functions of the algorithm alongside examples of its behaviour. A** illustrative atom-atom distance term of momentum target function showing torsion angle-mediated distance (black solid), and target glidepath (dark grey dashed). Minimisation reduces shaded area. **B** torsion angle trajectories are parameters, starting from linear interpolation (light grey dotted), addition of a first-order harmonic (dark grey dashed) and second-order (black solid). Relative contributions of perturbations serve as parameters. **C** after refinement of MalT transporter path, overall motion rendering the entire pathway from outward-open (orange) to inward-open (purple) in 1% steps. **D–F** zoom-in of residues 385 and 386, showing (**D**) start-, (**E**) end-state and (**F**) all 1% steps superimposed. **C–F** figures generated in PyMOL[64].

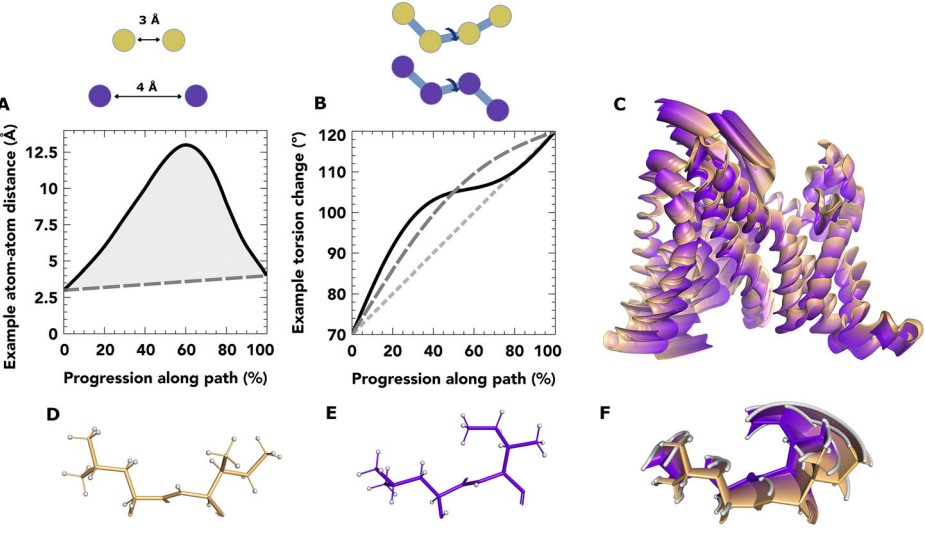

protein structure are considered rather than all pairs of atoms. Each pair has a starting and ending inter-atomic distance. A glidepath for these distances during the trajectory is sought that minimises the fluctuations in kinetic and potential energy of the whole protein during the transition (Fig. 2A). We have chosen a linear glidepath as the target. The justification for this is that for van der Waals and electrostatic energy terms, within the interatomic distances likely to be sampled at the beginning and end of the transition. Sampling distances in between these two values will minimise energy

variation required for the transition. We have also forbidden changes in bond lengths and angles, which would otherwise react poorly to a linear glidepath target. However, a perfect linear glide path is not achievable due to the constraints imposed by limited freedom of motion in torsion angle space. Therefore, the torsion angle motions are adjusted to minimise the differences between the actual interatomic distances and their ideal ones (Fig. 2B). Initially, torsion angles take a linear course between values from the start and end structure. Harmonic perturbations are added with

**Fig. 3 | Analysis of the transition for the EIIC domain of the MalT transporter. A** movements of the binding cavity in the EIIC domain, in 25% intervals along the trajectory with 0% and 100% corresponding to outward-open and inward-open respectively. In each figure the binding cavity, shown as a blue surface, was calculated using KVfinder[62]. **B** inside view of MalT showing movement of maltose (orange) in the binding cavity and opening of the inward pore. At 0% progression the maltose co-ordinates were taken directly from the outward open structure (PDB 5iws), for progressive frames the maltose was manually translated to fit inside the cavities using PyMOL[64]. **C** Close-up of the inside pore opening of MalT and movement of maltose.

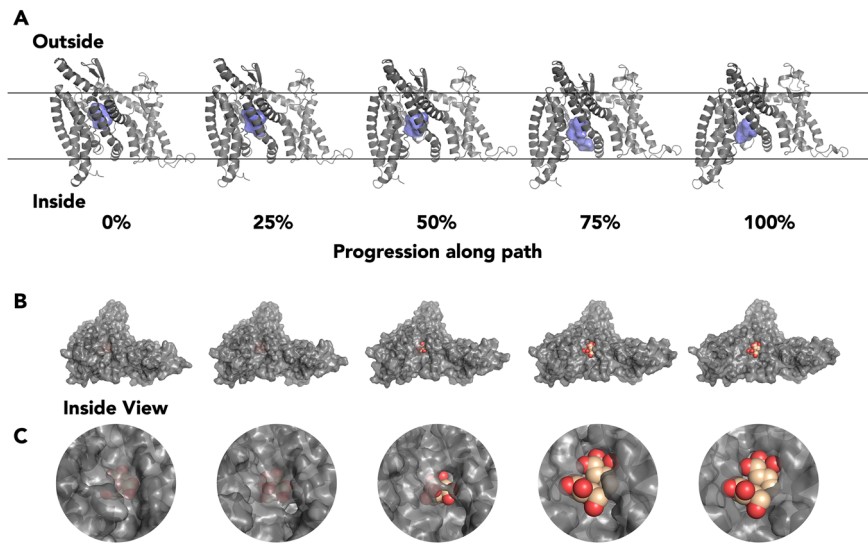

refinable amplitudes to minimise deviation from the ideal interatomic paths (called early-stage refinement, see "Methods"). Residual clashes in side-chains are then resolved by minimising a combined energy term (called late-stage refinement, see Methods).

## Elevator mechanism of maltose transport

The EIIC domain of MalT mediates the transport of maltose and is purported to act by the elevator subcategory of alternate access. The structure of this protein has been solved in an outward-open (PDB: 6bvg)[43] and inward-open (PDB: 5iws) conformation[44]. This was the simplest rearrangement in our study and we use this to demonstrate some characteristics of the method. We show the cold-inbetween path in Fig. 2C. Use of harmonic adjustments allows backbone atoms to follow near-perfect straight lines, despite moving by rotation around bonds. Sidechains, often undergoing much larger motions than the backbone, often take more curvaceous routes due to restrictions of that movement around a limited number of bonds (Fig. 2D–F).

The full path shows the coupled winding of alpha helix 2 (AH2) to become an extension of transmembrane helix 5 (TM5), and a subsequent collapse of the angle between AH2 and TM6 (Supplementary Data 1, Supplementary Movie 1). Despite the method not explicitly modelling a ligand, we found that it is nevertheless suitable for maintaining the appropriate size of the binding-cavity (Fig. 3A). Manual docking into the binding cavity demonstrates that the hydrogen-bonding pattern can be maintained at six equally spaced points along the pathway (Supplementary Fig. 3). This is because the starting and ending pairwise distances between atoms encode the size of the ligand-binding cavity, and there is no term in the target function which would lead to its collapse. Two residues V20 and V353 also appear to act as a gate, closing the maltose molecule off from the solvent during transit (Fig. 3B, C). The cavity is also maintained by miniature unwinding events in helix TM1, which forms one of the three walls of the elevator (Supplementary Movie 1).

## Rocking bundle mechanism of a LeuT fold protein

Next, we applied the algorithm to less straightforward rearrangements. NRAMP proteins transport metal ions in a wide array of organisms. In humans, mutations of NRAMP proteins lead to immune deficiency and anaemia. DraNramp is a tractable bacterial homologue of human NRAMP from *Deinococcus radiodurans* and a member of the ubiquitous LeuT superfamily[45,46]. DraNramp transports manganese and other heavy metal ions from outward- to inward-facing sides of the membrane. The first structure of DraNramp naturally crystallised in the inward-open form[47]. By introducing steric bulk in the outward-gate (G223W), DraNramp was crystallised in an outward-open form[48]. This mutation also impaired metal

transport and prevented N-ethylmaleimide (NEM) modification of an inward-gate reporter site (A53C), demonstrating inward-gate closure does not occur if the outward gate is forced open[48]. This therefore supports the alternate access mechanism. LeuT fold proteins are purported to work by the rocking-bundle subcategory of alternate access[49,50]. However, for all LeuT transporters, direct visualisation of the transition from inward- to outward-facing structures has been so far unattainable. Many solved structures of DraNramp have unmodelled loops. As a continuous chain of torsion angles is required for cold-inbetweening, we chose the fully-modelled Mn(II)-bound outward-open and inward-open conformations as start and end-points (PDBs: 8e6n and 8e6i respectively)[51]. To preserve the original sequence, A230 was mutated to wildtype methionine and manually altered in coot[52] to match the rotamer from the discontinuous apo- inward-open structure (PDB: 6d9w)[47]. We omitted the inward-occluded structure as it was unclear to what extent the G45R mutation was distorting the other-wise untethered TM1a helix.

In our pathway, a rapid and consistent contraction of TM10 and TM6a against TM1b closes off access to the ligand-binding site from the outward vestibule (Fig. 4A, Supplementary Data 2, Supplementary Movie 2). We monitor this using the area between the Cα atoms of G58, A227 and Q378 (outward triangle), which steadily collapses (Fig. 4B). The inward-facing vestibule is opened by expansion of a gap between TM1a, TM6b and TM8. The inward triangle for the inward-facing vestibule encompasses Cα atoms of S327, V233 and A53. In Fig. 4A, A53 rapidly moves away from the ligand-binding site through unwinding of the TM1a-b kinked region. However, this produces a shearing motion of TM1a against TM6b and TM8 in the first half of the pathway, which results in no overall change in the area of the outward triangle. It is only at the 50% mark that inward gate-opening picks up pace to allow solvent access to the ligand-binding site (Fig. 4C). This therefore supports previously anticipated behaviour and provides structural details for this hypothesis. The pathway was heavily influenced by constraints on the substantial rearrangement of the TM6b-TM7 linker, which required manual resolution of a clash. Here, I245 could either flip clockwise or anti-clockwise by 180° to the other side of the backbone. Q246 must complete a counter turn either 90° in one direction or 270° in the other (Supplementary Movie 2). One of the two residues needs to turn through the interior of the protein while the other must turn towards the solvent, but only Q246 was able to achieve this without substantial, unresolvable clashes on the loop interior. Accommodation of the 270° turn led to a substantial mid-trajectory displacement of TM6b. TM6b moves 1.2 Å out of alignment mid-trajectory (Supplementary Fig. 1). This suggests the mechanical hypothesis supports previous NEM-labelling data which indicated that TM6b moves substantially during conformational switching[53].

**Fig. 4 | Analysis of the transition for the DraN-Ramp transporter. A** movements at the ligand-binding site of DraNramp (TM helices 3 and 8 removed for clarity) in 20% intervals along the trajectory, with 0% and 100% corresponding to outward-open and inward-open, respectively. In each figure, the left dotted triangle reports on the solvent accessibility to the inward gate, and the right dotted triangle reports on that of the outward gate. Transmembrane helices 1, 5, 6 and 10 are shown in yellow, transmembrane helices 3, 4, 8 and 9 are shown in blue. The remainder of the polypeptide chain is shown in grey. **B** area of inward triangle (black solid line) and outward triangle (grey dashed line) along path. **C** surface render from outward and inward sides for five points on path, with residues rendered as sticks in (**A**) coloured light blue.

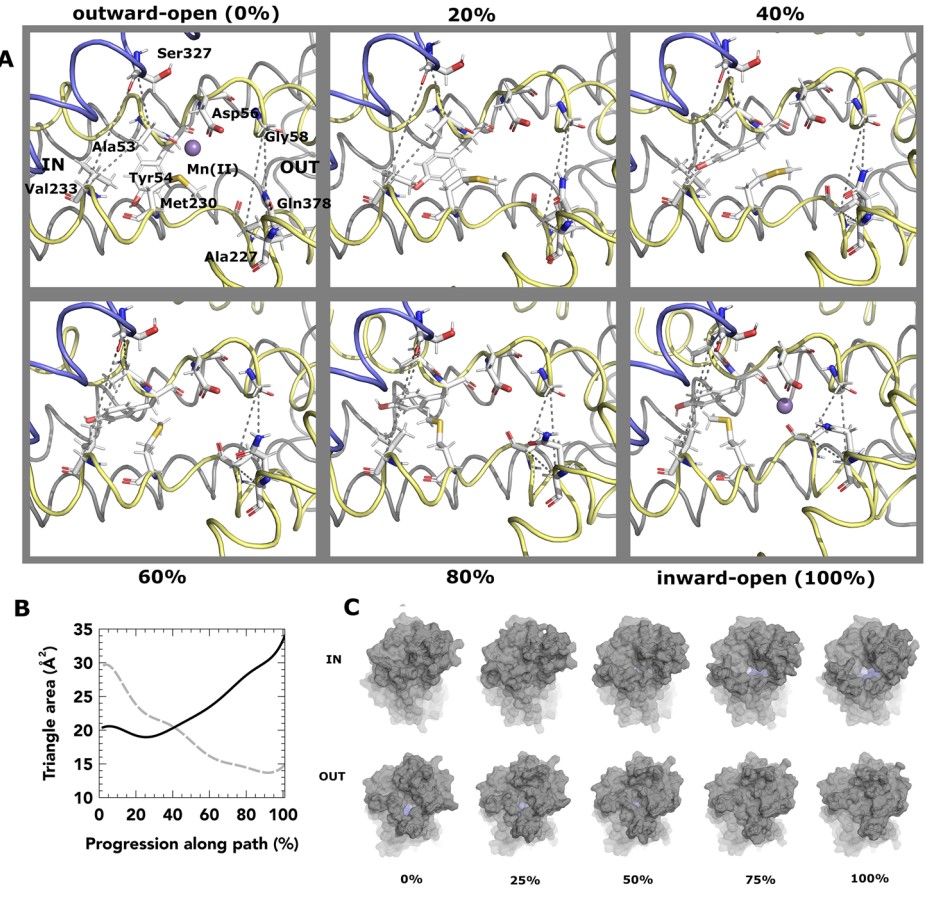

## Helix unwinding of the MATE transporter

Finally, we examined a member of the MOP superfamily, the multidrug and toxic compound extrusion (MATE) transporter[54]. MATE transports xenobiotics and metabolites across membranes of cells and organelles and in bacteria can confer resistance to antibiotics. MATE from *Pyrococcus furiosus* has a large 4300 Å$^3$ cavity. The inward-open and outward-open structures (PDBs: 6fhz and 6hfb)[55] differ by a dramatic rewinding of the central portion of the TM1 helix. Firstly, we checked Ramachandran outliers in the unwound structure: a peptide flip of G30 was found to fix a Ramachandran outlier in the inward-open conformation, which fit the electron density equally well. Other TM1 Ramachandran outliers were favoured in their fit to the electron density and considered to be realistic in light of the dynamics in this region. The helix, in its unwound state, bridges the large gap between the N-terminal domain and C-terminal domain; whereas the fully-wound helix is able to bridge the narrower gap in the outward-open state. One pertinent question is what combination of bond rotation directions allows the helix to rewind (or unwind) without causing internal steric clashes? By testing several combinations, we found a solution involving a mid-trajectory development and resolution of a kink in the middle of the helix (Fig. 5A, Supplementary Data 3, Supplementary Movie 3), which did not clash internally or with the rest of the protein (so-called "kinked winding"). This, however, protruded far from the helical axis. We also attempted a direct recoiling of the extended helix, in order to minimise off-axis motion and thereby conserve momentum (so-called "straight winding", Fig. 5B). However, we found this would be highly energetically unfavourable due to a close carbonyl-carbonyl clash in adjacent residues (M27 and M28) on the backbone. It also led to a 270° rotation of the N-terminal section of TM1 around its axis. Conversely, the kinked winding alternative has a more simple 180° shift of this region, which is a more achievable helix register slippage. We also note that at 95% progression through the kinked winding trajectory, the r.m.s.d. of $C_\alpha$ atoms of TM1 reaches a minimum against the

bent conformation[56], whereas no mid-trajectory agreement of the straight winding path has a smaller r.m.s.d. than the end-point, strongly suggesting straight winding is not the mechanism by which the alternate access model operates. The kinked winding protrusion extends into the cavity mid-trajectory and thereby blocks solvent access to the cavity (Fig. 5C) which does not occur for straight winding (Fig. 5D). We also checked for overlap of the kinked helix with ligand- and inhibitor-bound structures previously determined[56], and found that overlap with the ligand was insubstantial compared to that of the inhibitors (Supplementary Fig. 2).

## Discussion

Here we show that cold-inbetweening can be used to estimate the feasibility of particular transitions and their consequences. Although thermal motion is the only route by which molecular dynamics (and reality) can escape local energy minima, cold-inbetweening finds a common pathway around which true trajectories are likely to follow, albeit with additional noise attributed to heat. This, therefore, ought to complement the experimental determination of structures and simulations using the heat-driven engine of molecular dynamics simulations. Cold-inbetweening produces hypotheses for substantial transitions with a relatively low computational cost (Supplementary Table 1). These hypotheses could be evaluated experimentally, such as by comparison of mutants chosen to test them by structural determination or activity assays. The omission of thermal motion precludes direct comparison to the results of molecular dynamics simulations, and we do not assign a timescale to the trajectory. Although developed and demonstrated using experimentally determined end-point structures, cold-inbetweening may also be suitable for interpolation between MD-simulated frames. We show that for membrane transporter proteins, cold-inbetweening preserves ligand-binding states due to their inclusion in the start and end models even without explicitly modelling an energy term for electrostatics and solvent reorganisation. For this same reason, the consequences of omitting explicit

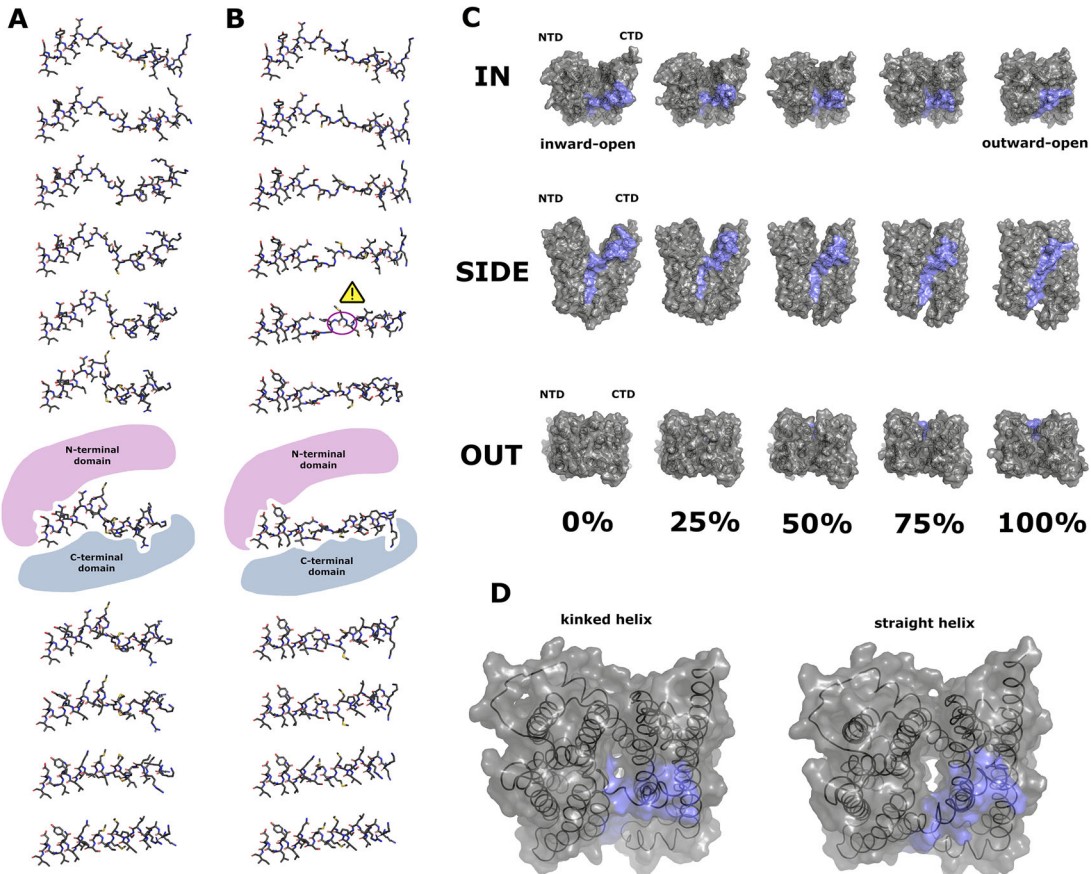

**Fig. 5 | Analysis of the transition for the MATE transporter. A** 10% increments in progression from 0% (top) to 100% (bottom) showing atomic trajectory of TM1 in MATE with torsion angle flips assigned as necessary to avoid backbone clashes ("kinked helix" trajectory). N-terminus is to the right. **B** similar increments in progression showing atomic trajectory of helix minimising out-of-axis motion ("straight helix" trajectory). Carbonyl clash on backbone highlighted with a warning sign. For (**A**, **B**) the 60% progression point is augmented by an illustration of relative positions of N-terminal and C-terminal domains. **C** view of kinked helix trajectory from the inward, side and outward views. TM1 residues 1-43 are marked in blue. **D** comparative views of kinked and straight helix trajectories at 70% trajectory, showing the aberrant opening of a solvent channel when all other protein motions with a straight helix trajectory are considered.

solvent or membrane lipids from the transition is partially mitigated. However, further expansions of the algorithm to include non-covalently interacting molecules is planned, which will provide a mechanism for modelling the effects of other proteins and small molecules which are well-supported by the electron density. In cases where there are imperfections in the fiducial end-points, these are imprinted on the trajectory. Therefore, substantial care should be taken to check Ramachandran outliers and regions of poor geometry before applying this method. Similarly, if the experimental data do not adequately justify the input structures, the trajectories calculated can be particularly misleading. Care must be taken to prevent inaccurate conclusions being drawn from poorly modelled protein structures. In cases where structural data are available[57], we show that is it possible to improve the model and rectify false outliers prior to calculating the trajectory as applied here to MATE. By cold-inbetweening the trajectory of transitions between protein conformations, we can generate testable hypotheses as to how the scaffold of the protein allows these transitions to contribute to the protein's function.

## Methods

### Construction of parameter set

The set of torsion angles refers to all torsion angles in a protein required to recalculate the protein's conformation. Each torsion angle varies as a function of $p$, which varies from 0 to 1 along the trajectory. The starting and ending structures of the protein are fixed. At $p = 0$, each torsion angle $t_i$ in the structure is defined to have the starting value $\theta_i^0$, and at $p = 1$, the ending value $\theta_i^1$. Each of the $t_i$ values for $i = 0 \ldots n$, where $n$ represents the maximum

torsion angle index, is a linear sum of terms as a function of $p$. The first of these terms (order 0) is a linear interpolation as in Eq. (1) and has no parameters.

$$t_i^0(p) = \theta_i^0 + p(\theta_i^1 - \theta_i^0) \tag{1}$$

Additional terms are defined for the maximum order 3. Amplitudes $f_j$ for each $j = 0\ldots3$, are refineable parameters, for which the default value is zero.

$$t_i^j(p) = f_j sin(jp\pi) \text{ where } j = 1, 2, 3 \tag{2}$$

These are combined to produce an interpolation for each torsion angle with 3 refineable parameters,

$$t_i = \sum_{j=0}^{3} t_i^j \tag{3}$$

### Torsion angle direction choice

For each torsion angle, the direction, clockwise or anticlockwise, must be chosen, which is not necessarily evident from the start or end states. Torsion angles which differ by no more than 30° between start and end state are considered to move 30° rather than 330° in the other direction. Torsion angles above 30° may be "flipped", i.e. the angle shifted by either +360 or

−360°, chosen as opposite to the angle's sign. The choice of 30° is a compromise between computational cost and the probability of a torsion angle flipping. The energy required to move 330° is reasonably assumed to be prohibitively higher than 30°. Each torsion angle moving more than 30° is flipped in turn, and checked against the overall distance travelled by each atom to determine if the flip should be accepted or rejected. For all other torsion angles, an algorithm is employed to choose the more appropriate direction within the context of the rest of the molecule as described below.

Each atom has a starting position, $a_i^0$, and an ending position $a_i^1$. At a position $p$ along the trajectory, an atom has a linearly interpolated position $a_i^p$:

$$a_i^p = a_i^0 + p(a_i^1 - a_i^0) \quad (4)$$

Sets of four atoms associated with each torsion angle were assigned consecutive values of $a_i^p$ for values of $p = 0, 0.1, 0.2, \ldots, 1.0$. This $p$-spacing was chosen to minimise errors caused by under-sampling while not unnecessarily increasing computational complexity. For the first iteration, a reference torsion angle was calculated from these sets of four atoms. For subsequent iterations, iteration $n$, an updated torsion angle was similarly calculated, and either -360, 0 or 360 was added in order to minimise the difference with the reference torsion angle from iteration $n-1$. This torsion angle is then carried through for the next calculation for iteration $n+1$. The final torsion angle for $p = 1.0$ is taken to indicate the direction of travel for this atom.

**Construction of early-stage target function**

Each atom has a starting position and ending position, which are a function of the set of torsion angles denoted by $\theta_i^0$ and $\theta_i^1$ over all $i$. This means that every pair of non-hydrogen atoms $i$ and $j$ have a calculable distance $d_p^{ij}$ for a given value of $p$, corresponding to an invariant beginning and ending distance at $p = 0$ and $p = 1$, respectively. Every pair of non-hydrogen atoms for which either $d_0^{ij}$ or $d_1^{ij}$ is below a fixed limit chosen to balance coverage of close contacts and computational efficiency (default value of 8 Å), and which are not covalently connected by three or fewer intervening bonds, is included in the early-stage target function. Hydrogen atoms are excluded as most hydrogens are riding off the backbone and therefore increase computational cost without introducing any degrees of freedom to be refined. The lack of experimental supporting data for freely rotatable hydrogen atoms also means we have chosen not to handle inter-atomic distances until the late-stage target function.

$$J_{\text{early}} = \sqrt{\sum_i \sum_{j, j \neq i} \int_0^1 \omega_{ij}(d_q^{ij} - d_p^{ij})^2 \, dp} \quad \text{where} \quad d_q^{ij} = d_0^{ij} + p(d_1^{ij} - d_0^{ij})$$

$$\text{and} \quad \omega_{ij} = (1 + d_p^{ij})^{-1} \quad (5)$$

This therefore forms a least squares target function between the target atomic distances or glidepaths $d_q^{ij}$, which are a linear interpolation with respect to $p$, and the calculated atomic distances $d_p^{ij}$ from the set of torsion angles for each value of $p$. In practice, $J_{\text{early}}$ is numerically integrated with a number of steps which is set to 12 by default. The $\omega_{ij}$ term upweights close contacts and helps escape overlapping protein backbones.

**Construction of late-stage target function**

A separate late-stage target function is included as the early-stage term does not sufficiently penalise close clashes. The late-stage target function measures van der Waals contacts and torsion angle energies, calculated over the pairs of atoms (including hydrogens) similarly below a fixed limit, which may need to be larger than 8 Å depending on the scale of the conformational change.

$$J_{\text{late}} = J_{\text{VdW}} + J_{\text{torsion}} \quad (6)$$

$J_{VdW}$ consists of a discontinuous function measuring van der Waals contacts above those expected from the target atomic distances using the two-term Lennard-Jones potential. This consists of a reference van der Waals energy term $VdW_q(ij)$ for the interpolated distance, here $\epsilon$ is the potential well depth and a measure of the interaction strength[58] and $\sigma^i$ is the van der Waals radius for atom $i$ multiplied by 0.75.

$$\text{VdW}_q(i, j) = 4\epsilon\left(\left(\frac{\sigma^i}{d_q^{ij}}\right)^{12} - \left(\frac{\sigma^i}{d_q^{ij}}\right)^6\right) \quad (7)$$

and a van der Waals energy term $\text{VdW}_p(i, j)$ for the calculated distance,

$$\text{VdW}_p(i, j) = 4\epsilon\left(\left(\frac{\sigma^i}{d_p^{ij}}\right)^{12} - \left(\frac{\sigma^i}{d_p^{ij}}\right)^6\right) \quad (8)$$

The $\text{VdW}_q$ term is often positive for hydrogens involved in hydrogen-bonding arrangements which should not necessarily be disturbed during the transition. Therefore, it is used as a threshold, so only the positive difference between $\text{VdW}_p$ and $\text{VdW}_q$ is considered.

$$\text{VdW}(i, j) = \begin{cases} \text{VdW}_p(ij) - \text{VdW}_q(ij) & \text{if } \text{VdW}_p(ij) > \text{VdW}_q(ij), \\ 0 & \text{otherwise}. \end{cases} \quad (9)$$

The van der Waals component of the late-stage target function is calculated as follows.

$$J_{\text{VdW}} = \sum_i \sum_j^{j \neq i} \text{VdW}(i, j) \quad (10)$$

Consider the pair of atoms (A, B) for each bond in the structure. A torsion angle will connect four atoms in total. Each bond will be the centre of a number of related torsion angles. To generate a general torsion angle energy function which approximates the energies associated with eclipsed and staggered conformations, each torsion angle is considered in turn. We find the set of $n$ torsion angles centred on each bond. We then define the total torsion energy as a sum of each part,

$$J_{\text{torsion}} = \sum_{(A,B)} \sum_i^n E(t_i) \quad (11)$$

where $E(t_i)$ is the corresponding energy term of the $i$th torsion,

$$E(t_i) = z_T\left(1 + \left(\frac{t_i}{\sigma}\right)^2\right)^{-3} \quad \text{where} \quad z_T = \sqrt{z_{A'}^2 + z_{B'}^2} \quad (12)$$

In this bell-curve function, $z_{A'}$ and $z_{B'}$ are the atomic numbers of the terminal atoms of the torsion angle directly bonded to A and B, respectively. $\sigma$ denotes a bell curve width set to 60°.

$J_{\text{late}}$ is numerically integrated with the same 12 steps for path calculations, but with some adjustments in $J_{\text{VdW}}$. By nature of slicing of the pathway into intervals for numerical integration, sometimes atoms may pass extremely close to each other, but the point of minimum distance falls in between two numerically integrated intervals. This would create a way in which a minimisation method can easily mis-fit, in conjunction with a rapidly-changing function like $J_{\text{VdW}}$. Therefore, a modification to this approach was taken: for an atom moving between intervals $n$ and $n + 1$, $n - 1$ and $n + 2$

intervals were also used to generate a cubic spline in dimensions $x, y, z$ where the polynomial must pass through each of these atom positions. When comparing the trajectories of atoms $i$ and $j$, the minimum distance between these atoms between fractional values between $n$ and $n + 1$ are found and this value is taken as the interval's partial sum towards the estimation of the total integral.

## Minimisation protocols

Minimisation against the $J_{early}$ term is carried out using the L-BFGS algorithm[59]. Minimisation is considered converged if there is less than a $10^{-3}$ change in the $J_{early}$ term after a L-BFGS refinement cycle. Gradient descent is carried out on $f_1$ for the set of all torsion angles until convergence is reached; then on $f_1, f_2$ until convergence is reached, and likewise for parameters $f_1, f_2, f_3$.

Minimisation against $J_{late}$ is carried out using the Nelder-Mead simplex descent algorithm[60]. As each sidechain is almost independent from the rest of the protein, barring other local sidechains, simplex descents are run for each sidechain concurrently. Parameters $f_1$, $f_2$, $f_3$ are refined for torsion angles which affect the sidechain and do not affect the main chain with a step size of 10°. After this, residues are ranked in terms of descending $J_{late}$ terms, including all atoms contributing to that residue's side or main chain. This is followed by additional simplex descents for the five residues with the highest values of $J_{late}$ term. This two-stage process is repeated until the $J_{late}$ term does not decrease by more than 5%.

## Manual adjustments of torsion angles

Due to the subtlety of interactions between torsion angle directional choices, manual adjustment of a handful of parameters is required. Some backbone torsion angles must be manually adjusted, where many neighbouring bonds frequently exceed 90°, as this is easily confusable with a 270° rotation in the opposite direction. These decision-making processes were aided by the RoPE GUI[42] which visually displays atom trajectories and makes bond flipping and harmonic adjustment accessible to users. Harmonic adjustment was occasionally employed during Nelder-Mead minimisation in order to escape van der Waals "tangled" side chains, which are trapped in local, but not global, minima due to crossed bonds.

## Preparation of structures

Structures were downloaded from PDB-REDO[61] to regularise the use of software used to refine structures. Structures were re-hydrogenated using *phenix.reduce*. These structures were prepared for cold-inbetweening using the RoPE software package[42] which optimises torsion angles to fit the experimentally determined model[41]. Before cold-inbetweening, to ensure the simplest transition, the 180°-symmetric torsion angles of affected side chains (phenylalanine, tyrosine, aspartate, glutamate, arginine) were made identical to avoid unnecessary 180° flips. Due to the lower resolution (> 2.4 Å) of structures involved, this was also extended to residues which could be distinguished in the electron density only at higher resolution (asparagine, glutamine, histidine), under the presumption that the hydrogen-bonding environment will remain relatively similar. Cold-inbetweening is implemented in the RoPE software package.

## Cavity calculations

The calculation of the ligand cavity for EIIC domain of the MalT transporter was performed using the KVFinder web server[62]. The molecular surface of the protein was calculated using a 1.7 Å probe radius and the inaccessible regions were calculated using a 4 Å probe radius. The outer limits of the cavity were defined using a 2.4 Å removal distance. The cavity volume cutoff was set according to the molecular volume of maltose 241.54 Å³. The molecular volume of maltose was calculated with MoloVol[63], using a carbon atom radius of 1.77 Å and oxygen radius of 1.50 Å, with a probe of 1.2 Å, grid resolution 0.2 Å and optimisation depth 4.

## Data availability

All models referenced in this study are available via the Protein Data Bank. The models used are 6BVG, 5IWS, 8E6N, 8E6I, 6FHZ, 6HFB, 6D9W, 3VVP

and 3VVR. Ensemble models of the three transitions are included in the Supplementary Material.

## Code availability

The code for performing the cold-inbetweening is included in git commit 9da67bd of the RoPE software (https://github.com/helenginn/rope, https://doi.org/10.5281/zenodo.6958154). The RoPE GUI[42] accepts structural files in the PDB file format and is able to export paths as multiple single model PDB files or one ensemble PDB file.

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

## Acknowledgements

We thank David Stuart, Stephen McCarthy, Godfrey Beddard and Ilme Schlichting for comments on this paper and providing valuable feedback. H.M.G. is funded by the Helmholtz Association, grant VH-NG-19-02 (Helmholtz Young Investigator Group).

## Author contributions

B.A.Y. and H.M.G. did data analysis, generated figures, and wrote the manuscript. H.M.G. wrote the algorithm code.

## Funding

## Competing interests

The authors declare no competing interests.
