## [Transparent Peer Review file · Communications Chemistry]

Calculation of minimum energy pathways in transport proteins.

Corresponding Author: Professor Helen Ginn

Version 0:

Reviewer comments:

Reviewer #1

(Remarks to the Author)

Review comments:

This paper presents a method using torsion angles to create trajectories of possible transitions between protein conformational states. The torsion angle perspective is very unique here, and represents a valuable way of providing models for the difficult-to-analyze situation of understanding different molecular conformations. The performance of the method in the case of transmembrane proteins in terms of preserving ligand binding states provides strong support for its utility and for providing actionable information regarding state transitions.

Major comments:

1. The paper tackles a majorly important area by providing models of large transitions between conformational states, using the torsion angle space for generating models. Given the uniqueness of the modeling, I would have benefitted from seeing a short summary of other modeling options and what they are designed to do. In other words, a very brief summary of and comparison to existing modeling approaches for capturing large conformational changes would be helpful. I noted that the other tools cited (refs 1 – 5, in the introduction section) are quite dated as computational tools go. Is there no more recent work in this area? I presume the interpolation performed by these alternate approaches is simple linear interpolation of atom positions.
2. There is insufficient discussion of the impact of the starting input structures, in terms of their quality and whether they do necessarily represent true points along a trajectory of conformational motion.
3. The Results section (4) starts with a brief discussion of inputs needed for the model, but this should be expanded. The inputs are exactly what data in what format (type of PDB file, etc), and after what preprocessing? How is the regularization of bond lengths done? Are close pairs of atoms meaning with a certain number of bonds? Close in physical space?
4. The authors should provide brief justification as to why minimization of KE/PE fluctuations is done for the whole protein, rather than just a neighborhood of the bond or atom pair being modeled. Is this to capture long-range allosteric effects?
5. The methods section should provide explicit mention of the inputs (file types, etc, see comment 3) and what is output from the algorithm (file type/format, how visualized, etc).
6. I am concerned that the MalT protein appears to be functionally homodimeric, but is modeled using cold in-betweening as just the monomer. Are similar motions expected simultaneously in the other monomer, and is there any between-protein conformational shifts occurring? Can the method capture between-protein relative motions?
7. Please provide the color assignment of the TM helices in Fig 3A in the caption.
8. In the TM use cases, the analyses provided are very compelling. Could the authors describe more how they selected which areas of the protein to focus on, and how the particular information was extracted from the output of the algorithm?
9. I would have benefitted from a more detailed caption (or text description) of Supp Fig 1. Please describe what exactly is being shown, and how to interpret the graphic.
10. The notation used in the Methods Section 6 was at times difficult and cumbersome. For instance, the variable n is used

to represent more than one thing: the largest value of the index i on the torsion angles (which I think defines the number of increments of p between 0 and 1), the order of the amplitude parameter f_n taking values in {1,2,3}, and as index of the iteration number (corresponds to the p increment?). Clearer definition of each term as it arises would be helpful.

11. How was it determined how many values of p between initial and final states were to be used? Was this due to computational cost? Was it the same for each atom pair? It looks from section 6.2 (p 8) that increments of 0.1 are chosen between 0 and 1. What would be the impact of a more finely grained p , other than obviously higher computing cost and more points on the trajectory.

12. The algorithm mentioned in 6.2 for selecting direction for torsion angles $>30\text{deg}$ needs to be explained in more detail.

13. Provide a brief sentence in the methods section about why both an early-stage and a late-stage target function is needed. This is mentioned briefly at the top of p 4 (end of 1st paragraph of Section 4), but needs to be expanded upon in the methods.

14. The notation in Eq 5 needs amending. The $j \neq i$ in the upper bound of the inner summation should be moved to below the sigma. It is not an upper limit, but just a condition on j . You could choose the lower bound of summation to either be ' $j, j \neq i$ ' or choose ' $j \neq i$ ' below the sigma, either would be correct.

15. Please provide a brief explanation as to why H atoms are included in the late-stage but not the early-stage target functions.

16. The choice of the cubic spline in computing the late target function is sound, given the nice smoothness properties of the spline. But I was concerned that this extra step would add substantially to the computing burden. The authors should discuss this.

17. Information on the run times of the algorithm should be provided, at least for the use cases (and what kind of system was it run on). The authors claim it is computationally inexpensive, but no concrete information about this is given.

18. How were the output files converted into the visualizations (eg those from Fig 1c). Can other viz tools other than Pymol be used?

19. Describe how these tools can be used to supplement or interface with MD simulation trajectories.

Minor comments/typos:

1. The authors should provide a short sentence explaining why only transport proteins were modeled. Would the method only work on large systems that experience enormous deformations? Describe or speculate on what this method would look like on some different types of proteins other than TM transport ones.

2. This is a very minor point, but a sentence explaining the name of the algorithm would have been nice. I presume "cold" refers to the fact that this is not heat-driven modeling, and "in-betweening" is about the modeled intermediate conformations between end points.

3. On p 3, in the sentence following reference [9], perhaps 'interface the ligand' should be 'interface with the ligand'.

4. Ref 13 is incomplete and needs the author listed.

5. In section 4.1, only the citation for the outward facing conformation of MalT is given. There is also a reference for the 6bvg conformation, and it should be added. Also, consider using the new updated PDB nomenclature for the structures used.

6. The species name *Deinococcus radiodurans* is repeatedly misspelled in the manuscript.

7. The quotes in the "tangled" chains in section 6.6 in the last sentence need to be corrected in TeX. Use `` for the start quote, and '' for end quote. TeX will not automatically format double quotes.

M. L. Lynch

Reviewer #2

(Remarks to the Author)

The authors suggest a method for construct pathways for protein conformational transitions, addressing an interesting problem. What is at present still missing in the article is the relation to the state of the art in the MD field, and assessments of the reliability of the suggested method:

1. The state of the art in the computational field is not reflected. The authors state on page 2:

"There is a need for computational calculation of trajectories, for which previous efforts [1–5] have made incremental progress in this regard."

The problem I see here is: The refs. 1 to 5 are a small subfraction of previous work. I recommend to authors to include efforts in the MD simulation community to find conformational transitions, transition states, and transition state ensembles. There is a vast body of previous work in this direction, which makes it difficult to single out few references. There are enhanced sampling methods, elastic network approaches, there is Markov state modeling and transition path sampling. And there are rather many examples of protein conformational transitions in standard MD simulations, obtained on supercomputers like Anton, or on standard GPUs, on timescales of microsecond to milliseconds. Yes, sampling conformational transitions in MD simulations is still complex, and many transitions occur on second timescales. I understand that the authors here suggest a method that is rather fast and applicable to many (or all?) protein systems for which end-state structures are available. But the statement in the first sentence of the introduction "Neither molecular dynamics simulations nor experimental methods provide sufficient information about large transitions between conformational states in proteins to fully characterise their mechanisms" is simply incorrect in its generality.

2. For any protein system exhibiting conformational changes, there is a "true pathway" for the conformational transition, with a rate-limiting transition state, or an ensemble of parallel pathways (e.g. for protein folding) with an associated transition-state ensemble. Even for systems with a dominant transition pathway, rather than an ensemble of pathways, how is it possible to know whether the algorithm suggested by the authors provides the correct transition path? In other words, how is it possible to know whether the conformational transition in the systems considered by the authors are biologically relevant? How can the method be tested and verified? I think the authors need to address these questions.

Reviewer #3

(Remarks to the Author)

The authors, Yorke and Ginn, introduce a new method called the "cold-inbetweening" algorithm, which generates trajectories in torsion angle space to study transitions between pre-defined protein metastable states (obtained from experiments). They claim to address a gap in current molecular dynamics (MD) simulation methods, a point I will revisit below. The authors applied their approach to examine the alternate access model in three membrane transporter superfamilies.

The method itself was carefully developed and is properly described.

However, the authors did not demonstrate that this method is indeed necessary for the systems studied. The RMSD between the start and end states for each system is not provided. Furthermore, the visual representations of the modeled systems are of low quality, making it difficult to discern whether large-scale motions actually occur. Consequently, it's unclear if enhanced sampling is needed or if regular MD would suffice. The authors do not compare their new method to existing techniques such as regular MD, metadynamics, or umbrella sampling MD. As a result, the benefits of their cold-inbetweening approach were not adequately demonstrated.

Essentially, their new method is a sophisticated interpolation between start and end states. However, the biological relevance of the pathways obtained from this approach needs to be demonstrated. Since the method does not calculate realistic energies, it fails to provide information about thermodynamics and kinetics. Notably, while "Thermodynamics" is listed as a keyword, the term does not appear once in the manuscript. Such discrepancy between paper content and representation should not occur.

A further shortcoming of this method is that solvent molecules and the lipid environment of the membrane proteins are not included in the modeling, severely limiting the biological relevance of this approach.

The presentation of the results requires enhancement. Specifically, when introducing concepts such as the "alternate access mechanism," the authors should include explanatory graphics. Visual aids would significantly improve reader comprehension, ensuring that all readers are "on the same page" regarding these complex mechanisms.

The figures themselves also need improvements:

Fig. 3: The labels in panel A are hardly readable. Where did the ion go in the 20%,..., 80% plots? The plots in C look all the same in the printed paper.

Fig. 4: In panels A and B, it is not clear what helices are shown. The figures are also too small. In C, the conformational transition is visible but larger figures with proper labeling of the relevant structure parts would help.

In summary, while the authors have developed a novel method, they have not adequately demonstrated its usefulness compared to existing techniques. The manuscript, in its current form, would be more suitable for a specialized journal. However, before submission, significant improvements to the presentation of results are necessary. These enhancements would strengthen the paper's impact and clarify the method's potential contributions to the field.

Version 1:

Reviewer comments:

Reviewer #1

(Remarks to the Author)

The thoroughly revised manuscript is much improved. I particularly appreciated seeing the extensive and clearly written summary of alternate methods for modeling conformational changes that has been added to the introduction. This goes a long way to put the proposed methods into context.

I am very satisfied with the changes made by the authors to address my (and the other reviewers') concerns. The additional detail provided did a lot to clear up areas of the manuscript where I had lurking questions. The authors are to be commended for putting together such an extensive range of additional detail, it has added a lot of clarity.

I have no further comments, and did not catch any typos in this revised manuscript.

Additional comments added 09/24:

Regarding whether the proposed method generates biologically relevant pathways between start and end states, the authors describe well the intended interpretation of trajectories as useful hypotheses requiring further mechanistic verification. It might be helpful, however, to have some sense of *why* we would expect the 'torsion angle approach' to provide better potential models of the conformational shifts between start and end states, compared to other approaches available. A sentence providing this rationale would strengthen the argument that the proposed method is generating useful, interpretable results.

Regarding the concerns about lack of explicit modeling of the protein environment, either w models accounting for the membrane or for the solvent, many readers would be expected to have concerns about the role of the protein environment in the cold-inbetweening models. Providing a brief sentence about why this is less of a concern for cold-inbetweening would be helpful. Also, explaining that future expansions of the method are planned for using protein environment information when it is supported by density in the crystal structures would be helpful.

Reviewer #2

(Remarks to the Author)

The authors have addressed my comments. I recommend publication.

Reviewer 1

Major comments:

1. The paper tackles a majorly important area by providing models of large transitions between conformational states, using the torsion angle space for generating models. Given the uniqueness of the modeling, I would have benefitted from seeing a short summary of other modeling options and what they are designed to do. In other words, a very brief summary of and comparison to existing modeling approaches for capturing large conformational changes would be helpful. I noted that the other tools cited (refs 1 – 5, in the introduction section) are quite dated as computational tools go. Is there no more recent work in this area? I presume the interpolation performed by these alternate approaches is simple linear interpolation of atom positions.

We have significantly improved the introduction to summarise other approaches to modeling transitions, including a range of molecular dynamics based approaches and alternative methods.

2. There is insufficient discussion of the impact of the starting input structures, in terms of their quality and whether they do necessarily represent true points along a trajectory of conformational motion.

We have extended the section in the discussion on this topic. It now reads: "In cases where there are imperfections in the fiducial end-points, these are imprinted on the trajectory. Therefore, substantial care should be taken to check Ramachandran outliers and regions of poor geometry before applying this method. Similarly, if the experimental data do not adequately justify the input structures, the trajectories calculated can be particularly misleading. Care must be taken to prevent inaccurate conclusions being drawn from poorly modelled protein structures."

3. The Results section (4) starts with a brief discussion of inputs needed for the model, but this should be expanded. The inputs are exactly what data in what format (type of PDB file, etc), and after what preprocessing? How is the regularization of bond lengths done?

We have extended the existing text in section 4:

The algorithm assumes regularised bond lengths, angles and optimised torsion angles of both starting and ending structures which should be supplied as PDB files with explicitly modelled hydrogen atoms. Regularisation is automatically taken care of in the process of importing the structures in the RoPE GUI where cold-inbetweening is implemented.

Are close pairs of atoms meaning with a certain number of bonds? Close in physical space?

We include the qualifier: separated by at least three bonds, less than 8 Å and not part of an aromatic ring

4. The authors should provide brief justification as to why minimization of KE/PE fluctuations is done for the whole protein, rather than just a neighborhood of the bond or atom pair being modeled. Is this to capture long-range allosteric effects?

We have clarified the sentence pertaining to this: "Close pairs of atoms (less than 8 Å) in the protein structure are considered rather than all pairs of atoms." You are right that whole-protein atom pairs would need justification!

5. The methods section should provide explicit mention of the inputs (file types, etc, see comment 3) and what is output from the algorithm (file type/format, how visualized, etc).

In addition to the response to point 3, which partially addresses this, we have also added the line " The GUI also allows for export of paths in PDB format, editing and visualisation of the pathway using a draggable slider."

6. I am concerned that the MalT protein appears to be functionally homodimeric, but is modeled using cold in-betweening as just the monomer. Are similar motions expected simultaneously in the other monomer, and is there any between-protein conformational shifts occurring? Can the method capture between-protein relative motions?

Sharing partially in response to Ref #3, the MalT pathway shows a monomer. MalT is dimeric, but the channel and dynamic regions were far enough from the interface, so we considered it an acceptable candidate for demonstration of pathway calculations using the software in its current form. However, we cannot make any assumptions about the protein-protein interactions due to the

limitations in the current implementation. We agree that considering dynamics across protein interfaces is important and are looking forward to including non-covalent interactions in the future. The description of an internal coordinate system to make this work properly is significantly more complex and outside the scope for this paper.

7. Please provide the color assignment of the TM helices in Fig 3A in the caption.

We have added this information to the figure legend.

8. In the TM use cases, the analyses provided are very compelling. Could the authors describe more how they selected which areas of the protein to focus on, and how the particular information was extracted from the output of the algorithm?

Sorry, we aren't entirely certain which use case is in question here. If referring to the movement of helices in DraNramp, the linker in the TM6b-TM7 region was highlighted by the algorithm itself due to the manual resolution of a complex clash. This was then cross-referenced to the literature as TM6 is considered a key conveyor of protein motion. We have tweaked the text slightly to indicate the logic involved. It now reads: "The pathway was heavily influenced by constraints on the substantial rearrangement of the TM6b-TM7 linker, which required manual resolution of a clash." We hope we have identified the correct point.

9. I would have benefitted from a more detailed caption (or text description) of Supp Fig 1. Please describe what exactly is being shown, and how to interpret the graphic.

We have tried to provide a clearer description of Supp. Fig 1.

10. The notation used in the Methods Section 6 was at times difficult and cumbersome. For instance, the variable n is used to represent more than one thing: the largest value of the index i on the torsion angles (which I think defines the number of increments of p between 0 and 1), the order of the amplitude parameter f_n taking values in $\{1,2,3\}$, and as index of the iteration number (corresponds to the p increment?). Clearer definition of each term as it arises would be helpful.

We have altered the notation so n is only ever used to denote a maximum index, and now i and j are iterated over the sum. We have also removed the duplicate term, summing $j=0$ twice in Eq. 3. Hopefully it is clearer now.

11. How was it determined how many values of p between initial and final states were to be used? Was this due to computational cost? Was it the same for each atom pair? It looks from section 6.2 (p 8) that increments of 0.1 are chosen between 0 and 1. What would be the impact of a more finely grained p , other than obviously higher computing cost and more points on the trajectory.

Yes - we chose it based on the following principle: "This p -spacing was chosen to minimise errors caused by under-sampling while not unnecessarily increasing computational complexity." which we have added to the text.

12. The algorithm mentioned in 6.2 for selecting direction for torsion angles $>30^\circ$ needs to be explained in more detail.

We have altered the text: For each torsion angle, the direction of clockwise or anticlockwise must be chosen, which is not necessarily evident from the start or end states. Torsion angles which differ by no more than 30° between start and end state are considered to move 30° rather than 330° . Torsion angles above 30° may be "flipped", i.e. the angle shifted by either $+360^\circ$ or -360° , chosen as opposite to the angle's sign. The choice of 30° is a compromise between computational cost and the probability of a torsion angle flipping. The energy required to move 330° is reasonably assumed to be prohibitively higher than 30° . Each torsion angle moving more than 30° is flipped in turn, and checked against the overall distance travelled by each atom to determine if the flip should be accepted or rejected. For all other torsion angles, an algorithm is employed to choose the more appropriate direction within the context of the rest of the molecule as described below.

13. Provide a brief sentence in the methods section about why both an early-stage and a late-stage target function is needed. This is mentioned briefly at the top of p 4 (end of 1st paragraph of Section 4), but needs to be expanded upon in the methods.

We have added the line: "A separate late-stage target function is included as the early-stage term does not sufficiently penalise close clashes." to the methods section.

14. The notation in Eq 5 needs amending. The $j \neq i$ in the upper bound of the inner summation should be moved to below the sigma. It is not an upper limit, but just a condition on j . You could choose the lower bound of summation to either be ' $j, j \neq i$ ' or choose ' $j \neq i$ ' below the sigma, either would be correct.

Fixed! Thank you.

15. Please provide a brief explanation as to why H atoms are included in the late-stage but not the early-stage target functions.

We have added to the early-stage text, "Hydrogen atoms are excluded as most hydrogens are riding off the backbone and therefore increase computational cost without introducing any degrees of freedom to be refined. The lack of experimental supporting data for freely rotatable hydrogen atoms also means we have chosen not to handle inter-atomic distances until the late-stage target function."

16. The choice of the cubic spline in computing the late target function is sound, given the nice smoothness properties of the spline. But I was concerned that this extra step would add substantially to the computing burden. The authors should discuss this.

There is certainly an initial impact in computational cost of the cubic spline to identify clashes. What we found was that (a) this was mitigated quite substantially by multi-threading the calculation, where each p-interval could be independently assessed once the necessary neighbouring calculations were ready, and (b) the reduction in the necessity for resolving manual clashes offset the increased computational cost in the stages without human intervention, and reduced the burden on the user.

17. Information on the run times of the algorithm should be provided, at least for the use cases (and what kind of system was it run on). The authors claim it is computationally inexpensive, but no concrete information about this is given.

We were a little reluctant to include concrete information because the speed is dependent on the experience of the user, who does need to provide some guidance for the trajectory calculations in more complex scenarios like these in the paper. Since this process is interactive and depends on user intervention to

resolve difficult clashes, these times are not purely computational work. However, one author has now repeated the trajectory calculations with a stopwatch. We have included this in a supplementary table with the appropriate caveat that the times (ranging from 7 to 35 minutes) were carried out by an experienced user and that inexperienced users may not achieve this straight away. Nevertheless, inclusion of the timescales shows that they can be carried out in less than an hour on a decent laptop, so we agree that this is an important ballpark figure to include in the paper. Thank you for the suggestion.

18. How were the output files converted into the visualizations (eg those from Fig 1c). Can other viz tools other than Pymol be used?

We have added this information to the data availability section.

19. Describe how these tools can be used to supplement or interface with MD simulation trajectories.

It is out of scope of the paper to provide a rigorous description of combining MD and cold-inbetweening as this would need to be validated. However we do see value in using cold-inbetweening to interpolate between MD frames, and we have added a light-touch comment in the discussion: "Although developed and demonstrated using experimentally determined end-point structures, cold-inbetweening may also be suitable for interpolation between MD-simulated frames."

Minor comments/typos:

1. The authors should provide a short sentence explaining why only transport proteins were modeled. Would the method only work on large systems that experience enormous deformations? Describe or speculate on what this method would look like on some different types of proteins other than TM transport ones.

We have added a point in the introduction to address this. We focused on membrane proteins as these represent high complexity MD calculations typically underserved by computational techniques.

2. This is a very minor point, but a sentence explaining the name of the algorithm would have been nice. I presume "cold" refers to the fact that this is

not heat-driven modeling, and 'in-betweening' is about the modeled intermediate conformations between end points.

These presumptions are correct and we have added a short explanation of the name to the introduction.

3. On p 3, in the sentence following reference [9], perhaps 'interface the ligand' should be 'interface with the ligand'.

We have corrected this.

4. Ref 13 is incomplete and needs the author listed.

Thank you, we have corrected the reference.

5. In section 4.1, only the citation for the outward facing conformation of MaT is given. There is also a reference for the 6bvg conformation, and it should be added. Also, consider using the new updated PDB nomenclature for the structures used.

We have added this reference.

6. The species name *Deinococcus radiodurans* is repeatedly misspelled in the manuscript.

Thank you, this is fixed.

7. The quotes in the "tangled" chains in second 6.6 in the last sentence need to be corrected in TeX. Use `` for the start quote, and '' for end quote. TeX will not automatically format double quotes.

Thank you and thank you very much for a thorough review.

Reviewer #2 (Remarks to the Author):

The authors suggest a method for construct pathways for protein conformational transitions, addressing an interesting problem. What is at present still missing in the article is the relation to the state of the art in the MD field, and assessments of the reliability of the suggested method:

1. The state of the art in the computational field is not reflected. The authors state on page 2:

"There is a need for computational calculation of trajectories, for which previous efforts [1–5] have made incremental progress in this regard."

The problem I see here is: The refs. 1 to 5 are a small subfraction of previous work. I recommend to authors to include efforts in the MD simulation community to find conformational transitions, transition states, and transition state ensembles. There is a vast body of previous work in this direction, which makes it difficult to single out few references. There are enhanced sampling methods, elastic network approaches, there is Markov state modeling and transition path sampling. And there are rather many examples of protein conformational transitions in standard MD simulations, obtained on supercomputers like Anton, or on standard GPUs, on timescales of microsecond to milliseconds. Yes, sampling conformational transitions in MD simulations is still complex, and many transitions occur on second timescales. I understand that the authors here suggest a method that is rather fast and applicable to many (or all?) protein systems for which end-state structures are available. But the statement in the first sentence of the introduction "Neither molecular dynamics simulations nor experimental methods provide sufficient information about large transitions between conformational states in proteins to fully characterise their mechanisms" is simply incorrect in its generality.

We agree with the points raised and have significantly improved the introduction to include a summary of molecular dynamics based approaches suitable for sampling conformational transitions. We have removed the statement "Neither molecular dynamics simulations nor experimental methods provide sufficient information about large transitions between conformational states in proteins to fully characterise their mechanisms" and reworded this section to better reflect the current state-of-the-art and to contextualise the cold-inbetweening approach.

2. For any protein system exhibiting conformational changes, there is a "true pathway" for the conformational transition, with a rate-limiting transition state, or an ensemble of parallel pathways (e.g. for protein folding) with an associated transition-state ensemble. Even for systems with a dominant transition pathway, rather than an ensemble of pathways, how is it possible to know whether the algorithm suggested by the authors provides the correct transition path? In other words, how is it possible to know whether the conformational transition in the systems considered by the authors are biologically relevant? How can the method be tested and verified? I think the authors need to address these questions.

We consider the gold standard to be comparison to experiment, e.g. testing of hypotheses by mutagenesis followed by either structure determination or activity assays. We have altered the text to guide the reader as to how we expect the generated hypotheses to be evaluated. "These hypotheses could be evaluated experimentally, such as by comparison of mutants chosen to test them by structural determination or activity assays."

Reviewer #3 (Remarks to the Author):

The authors, Yorke and Ginn, introduce a new method called the "cold-inbetweening" algorithm, which generates trajectories in torsion angle space to study transitions between pre-defined protein metastable states (obtained from experiments). They claim to address a gap in current molecular dynamics (MD) simulation methods, a point I will revisit below. The authors applied their approach to examine the alternate access model in three membrane transporter superfamilies.

The method itself was carefully developed and is properly described.

However, the authors did not demonstrate that this method is indeed necessary for the systems studied. The RMSD between the start and end states for each system is not provided.

The RMSD between the start and end structures of each exemplar system are now provided in Supplementary Table 1.

Furthermore, the visual representations of the modeled systems are of low quality, making it difficult to discern whether large-scale motions actually

occur. Consequently, it's unclear if enhanced sampling is needed or if regular MD would suffice.

Unfortunately the video submissions were limited to 10 MB for the initial submission. Although we tried to encode the videos without visual distortion, this may still have impacted the overall resolution of the videos. The master videos are three times larger which we hope to upload for the final submission.

The authors do not compare their new method to existing techniques such as regular MD, metadynamics, or umbrella sampling MD. As a result, the benefits of their cold-inbetweening approach were not adequately demonstrated.

We have extended our introduction to provide a broader overview of MD-based methods. In point 17 of reviewer #1's request, we include some information on computational run-time.

Essentially, their new method is a sophisticated interpolation between start and end states. However, the biological relevance of the pathways obtained from this approach needs to be demonstrated.

It is notoriously difficult to experimentally observe transitions between low-energy conformations and has not been done for such large transitions. As we have no direct comparison, we provide circumstantial evidence that the trajectories that we generate are compatible for all described proteins with the alternate access model. However we have described the tool throughout the text in terms of "hypothesis generation" to make it clear to potential users that these paths must be taken with a healthy dose of skepticism, as should any molecular dynamics trajectory as well.

Since the method does not calculate realistic energies, it fails to provide information about thermodynamics and kinetics. Notably, while "Thermodynamics" is listed as a keyword, the term does not appear once in the manuscript. Such discrepancy between paper content and representation should not occur.

You are right, we have removed that keyword from the manuscript.

A further shortcoming of this method is that solvent molecules and the lipid environment of the membrane proteins are not included in the modeling, severely limiting the biological relevance of this approach.

There are two aspects to this: one is (a) the interface between multiple molecules, which is not covered in this paper; the other is (b) the lack of modelling for either bulk solvent or membrane. In molecular dynamics, neglecting to model either the solvent or membrane (or even an implicit solvent) would be fatal for the trajectory results.

The method provides useful information without the membrane being explicitly modelled, whereas explicitly modelling membranes in MD simulations is complex due to the difficulty of generating realistic membrane models and forcefields as well as the additional computational cost.

Including non-covalent interactions as described in (a) is in development. The current paper reports internal dynamics of single monomers only. We are very excited to eventually offer cross-protein interactions, but the torsion angle system doesn't automatically transfer to non-covalent interactions. It is however extendable by defining a different internal coordinate system. This is of significantly higher complexity and is out of scope for the current paper. For this reason the manganese ion in the DraNrap structure is not shown in intermediate states in figure 3. Similarly, the MalT pathway shows a monomer. MalT is dimeric, but the channel and dynamic regions were far enough from the interface, so we considered it an acceptable candidate for demonstration of pathway calculations using the software in its current form. We do not speculate on the movement of the protein within the double membrane.

For (b) the detrimental effect of not including these is not as extreme as it would be if a molecular dynamics trajectory were to model a membrane protein in vacuum. However, we would like to model explicit waters and lipids as seen in the crystal structures in the future once the algorithm is extended. Cold-inbetweening provides flexibility for parametrising hybrid explicit/implicit solvent models and this will be also very important for imposing the effect of explicit water on protein motion as well.

The presentation of the results requires enhancement. Specifically, when introducing concepts such as the "alternate access mechanism," the authors should include explanatory graphics. Visual aids would significantly improve reader comprehension, ensuring that all readers are "on the same page" regarding these complex mechanisms.

We have expanded the description of all the alternate access mechanisms in the introduction to make it more accessible to a general audience, and included a visual aid and the connections to each of our examples as the first figure. Thank you for the suggestion.

The figures themselves also need improvements:

Fig. 3: The labels in panel A are hardly readable. Where did the ion go in the 20%,..., 80% plots? The plots in C look all the same in the printed paper.

We have increased the size of the labels in panel A and have changed the colours used in panel C to highlight differences along the pathway. The ion is not included in the intermediate structures. We describe why in the previous point.

Fig. 4: In panels A and B, it is not clear what helices are shown. The figures are also too small. In C, the conformational transition is visible but larger figures with proper labeling of the relevant structure parts would help.

The figure size has been increased and we have added NTD and CTD labels to panel C. The helix shown in blue in panels A and B is already described as TM1 in the figure legend.

In summary, while the authors have developed a novel method, they have not adequately demonstrated its usefulness compared to existing techniques. The manuscript, in its current form, would be more suitable for a specialized journal. However, before submission, significant improvements to the presentation of results are necessary. These enhancements would strengthen the paper's impact and clarify the method's potential contributions to the field.

We thank you for your helpful review.

We would like to thank the referees for their help in improving the manuscript.

REVIEWERS' COMMENTS:

Reviewer #1 (Remarks to the Author):

The thoroughly revised manuscript is much improved. I particularly appreciated seeing the extensive and clearly written summary of alternate methods for modeling conformational changes that has been added to the introduction. This goes a long way to put the proposed methods into context.

I am very satisfied with the changes made by the authors to address my (and the other reviewers') concerns. The additional detail provided did a lot to clear up areas of the manuscript where I had lurking questions. The authors are to be commended for putting together such an extensive range of additional detail, it has added a lot of clarity.

I have no further comments, and did not catch any typos in this revised manuscript.

We thank reviewer 1 for their comments and suggestions in the previous round of revision and we are pleased we have addressed all of these.

Additional comments added 09/24:

Regarding whether the proposed method generates biologically relevant pathways between start and end states, the authors describe well the intended interpretation of trajectories as useful hypotheses requiring further mechanistic verification. It might be helpful, however, to have some sense of *why* we would expect the 'torsion angle approach' to provide better potential models of the conformational shifts between start and end states, compared to other approaches available. A sentence providing this rationale would strengthen the argument that the proposed method is generating useful, interpretable results.

We have slightly modified the introduction to explicitly state the rationale to strengthen our argument.

The algorithm mimics the nature of protein flexibility by allowing rotation around bonds. According to the equipartition theorem, energy terms should be equally distributed between bond stretching, angle bending and rotations. However, due to the much higher magnitude of forces involved in altering bond lengths and angles in comparison to rotations around the bond, the torsion angle changes are far more significant for large conformational changes in protein structure. Therefore, due to the simplification of the parameter space to include only torsion angles, larger conformational changes can be studied at a lower computational cost than other methods.

Regarding the concerns about lack of explicit modeling of the protein environment, either w models accounting for the membrane or for the solvent,

many readers would be expected to have concerns about the role of the protein environment in the cold-inbetweening models. Providing a brief sentence about why this is less of a concern for cold-inbetweening would be helpful. Also, explaining that future expansions of the method are planned for using protein environment information when it is supported by density in the crystal structures would be helpful.

We agree and have added the following to the discussion:

We show that for membrane transporter proteins, cold-inbetweening preserves ligand-binding states due to their inclusion in the start and end models even without explicitly modelling an energy term for electrostatics and solvent reorganisation. For this same reason, the consequences of omitting explicit solvent or membrane lipids from the transition is partially mitigated. However, further expansions of the algorithm to include non-covalently interacting molecules is planned, which will provide a mechanism for modelling the effects of other proteins and small molecules which are well-supported by the electron density.

Reviewer #2 (Remarks to the Author):

The authors have addressed my comments. I recommend publication.

We thank reviewer 2 for their recommendation to publish.